# Changes in Mass Treatment of the Canine Parvovirus ICU Population in Relation to Public Policy Changes during the COVID-19 Pandemic

**DOI:** 10.3390/v12121419

**Published:** 2020-12-10

**Authors:** Kevin Horecka, Nipuni Ratnayaka, Elizabeth A. Davis

**Affiliations:** Research Department, Austin Pets Alive!, Austin, TX 78703, USA; nipuni.ratnayaka@americanpetsalive.org (N.R.); elizabethdavis7@gmail.com (E.A.D.)

**Keywords:** canine, canine parvovirus, COVID-19, veterinary epidemiology

## Abstract

Previous work has indicated that canine parvovirus (CPV) prevalence in the Central Texas region may follow yearly, periodic patterns. The peak in CPV infection rates occurs during the summer months of May and June, marking a distinct “CPV season”. We hypothesized that human activity contributes to these seasonal changes in CPV infections. The COVID-19 pandemic resulted in drastic changes in human behavior which happened to synchronize with the CPV season in Central Texas, providing a unique opportunity with which to assess whether these society-level behavioral changes result in appreciable changes in CPV patient populations in the largest CPV treatment facility in Texas. In this work, we examine the population of CPV-infected patients at a large, dedicated CPV treatment clinic in Texas (having treated more than 5000 CPV-positive dogs in the last decade) and demonstrate that societal–behavioral changes due to COVID-19 were associated with a drastic reduction in CPV infections. This reduction occurred precisely when CPV season would typically begin, during the period immediately following state-wide “reopening” of business and facilities, resulting in a change in the typical CPV season when compared with previous years. These results provide evidence that changes in human activity may, in some way, contribute to changes in rates of CPV infection in the Central Texas region.

## 1. Introduction

The canine parvovirus (CPV) is a highly contagious gastrointestinal virus which, without treatment, has a fatality rate as high as 90% among domestic dogs [1,2]. With treatment, the fatality rate improves to 14% [3], but is still among the most common infectious diseases in dogs with high morbidity and mortality [4]. The disease incubates for approximately five days [5]. Following symptom onset, CPV takes a median of 11 days to resolve [3] if the animal survives the infection. Because of the significant risk that CPV presents to domestic dog populations, understanding factors influencing CPV incidence is of critical importance to reduce disease transmission in unvaccinated populations.

Seasonal effects in CPV have been observed worldwide [6,7,8,9,10]. In Central Texas, seasonal trends in incidence have been observed during the past decade, specifically peaking in the late spring/early summer months of May and June. Although much is known about the mode of transmission (i.e., viral spread via the fecal–oral route [11]) and it is standard practice in Central Texas animal shelters to inoculate against for CPV using the highly effective and widely available CPV vaccine, significant natural disease reservoirs must persist to perpetuate the disease among domesticated dogs. These reservoirs may exist in the form of wildlife (big cats, racoons, racoon dogs, arctic foxes, and mink; though evidence of transmission via this route is unclear [12,13,14]) and/or unvaccinated feral animal populations. Additionally, surfaces in the environment such as grass or pavement may house infections, given CPV’s ability to survive on surfaces for extended periods of time due to the hardy nature of this nonenveloped virus [4,15]. Previous work [3] has suggested that human activity may be, in some way, driving CPV infections. Thus, the drastic changes in human behavior during and immediately following the “stay-at-home” orders due to the coronavirus disease 2019 (COVID-19) pandemic provided a unique opportunity to gather further evidence with regard to this hypothesis.

The COVID-19 pandemic, caused by the severe acute respiratory syndrome coronavirus 2 (SARS-CoV-2) pathogen, resulted in major policy changes across the state of Texas, the United States, and much of the world. In Texas’s Travis County, such policy changes included non-essential business closures [16,17,18], school closures [19,20], and stay-home orders [18,21], designed to mitigate COVID-19 transmission in an effort to avoid exceeding healthcare capacity. Mobility and navigation request data from cellular phones [22,23], consumer activity data [24], and transportation department traffic density maps [25] generally indicated that there were substantial reductions in extra-household human activities while stay-home orders were in effect (and immediately after orders were lifted). Although good quality data on how outdoor recreational and physical activities (e.g., hiking, camping, walking dogs, etc.) were impacted are sparse, it is possible that the initial panic or concern surrounding the pandemic, in conjunction with closures of some outdoor venues (i.e., parks, bars, dog parks, beaches, etc.), discouraged people from partaking in outdoor activities as frequently as usual, in particular earlier in the shutdown period. Changes in human behavior during and shortly after the “stay-at-home” period may have, subsequently, impacted the incidence of CPV infection compared with previous years by reducing the frequency, duration, and intensity of contact between owned dogs and CPV infection sources.

Austin Pets Alive! (APA!), a non-profit, closed admissions animal shelter in Austin, TX (Travis County) has maintained a dedicated intensive care unit (ICU) for the exclusive treatment of CPV-infected dogs since 2008, having treated over 5000 animals during that period [3]. Moreover, as the organization has evolved, data collection has substantially increased in resolution, from monthly aggregate population numbers only to individual animal care records beginning reliably in 2017. A previous investigation used these data sources to assess patient signalment, disease trajectory and prognosis, and seasonal trends in disease incidence [3]. In the present study, we use more recent data from APA! in order to examine changes in population in the ICU during the COVID-19 pandemic in conjunction with changes in policy which may have impacted human activities and behaviors contributing to the disease spread. We present several hypotheses that might further elucidate the relationship between human activity and CPV infection and, therefore, may lead to policy and protocol changes which can reduce the spread of CPV. Through this observational study, this work seeks to investigate whether human activity, marked by public policy-driven behavioral changes during the COVID-19 pandemic, is associated with CPV spread in the Central Texas region.

## 2. Materials and Methods

### 2.1. Data Sourcing

In order to assess population levels of CPV-infected animals over time, we examined data collected via our end-of-shift ICU report. These data include the number of animals in the ICU measured twice daily, once in the morning and once in the evening (see [3] for a detailed prior analysis of pre-COVID-19 elements of these data). Data on COVID-19 policies were collected from the Austin City Government website [18]. Information on Austin Animal Center and Austin Pets Alive! (the two primary shelters involved in this study) and their associated intake policies was collected directly from the shelter administrators at each site. Intake source location data were collected via the intake paperwork for each animal during the time periods in question (note that data of this granularity were only available from 2017 onward, while the population data being examined were available from 2013 and onward). Finally, monthly aggregate counts of at-home treatment for CPV patients (see [3] for details on when this protocol was implemented) were observed as a possible explanatory variable for changes in CPV populations in the ICU.

### 2.2. Shelter Intake Policy and Geographic Sourcing

Austin Pets Alive!’s Parvo ICU intake practices did not change during any of the periods in question (i.e., no artificial reduction in population due to policy was present). In addition, no changes in geographic distribution of animal intake sources at APA! were observed, as measured by a McNemar–Bowker test for multiple correlation proportions (2017–2019 vs. 2020; χ^2^ = 9.48, ν = 528, p~1).

### 2.3. Statistical Analyses

To analyze the difference in CPV infections between 2020 and prior years, statistical tests were performed on the difference between the nadir of CPV infections in 2020 compared with the remainder of CPV infections in prior years. This allowed us to assess the likelihood of such an extreme difference occurring by chance in the model. Because the distribution of differences in CPV infections between 2020 and prior years was not normal and not easily correctable to normal (via boxcox, sqrt, log, or other common transformations), we were unable to assess a z-score or other measures of likelihood for this observed difference using parametric methods. As a result, a non-parametric, Gaussian kernel density estimate over the distribution of differences was computed with a bandwidth parameter determined by Scott’s factor (*w* = 0.33, in this case; for all comparisons, [26]).

Per previous methods of analyzing these types of data [3], a two-week moving average was used on all daily-reported time-series data. This was carried out primarily to reduce the presence of temporal autocorrelation due to animals being measured on recurring days, given that the disease time course is approximately 11 days. All statistical tests used significance thresholds of *p* < 0.05.

## 3. Results

Differences in CPV Infections during the COVID-19 Pandemic

In order to assess differences in CPV infections during the COVID-19 pandemic compared with previous years (raw data visualized in Figure 1), two primary comparisons were performed. First, a comparison of 2019 and 2020 CPV infections over time was conducted to assess the direct year-over-year difference which may have been due to the pandemic (Figure 2A). Second, a comparison of the average of the 2013–2019 CPV infections over time and the 2020 CPV infections over time was also conducted (Figure 2B).

A distinct dip in CPV ICU population can be visually observed during the “stay-at-home” period (defined as the time period between the City of Austin “stay-at-home” order and the state-wide “reopening” order; see Figure 1 and Figure 2B). However, a nearly identical dip was present in 2019 as well (see Figure 1), suggesting that this dip cannot be easily associated with the pandemic. However, immediately following the state-wide “reopening” event, results from the kernel density estimate analysis show that the decrease in CPV ICU population after the “stay-at-home” period is statistically significant when compared with the differences throughout the rest of the year with prior years (Figure 2A [2019 vs. 2020; *p* = 0.007] and Figure 2B [2013–2019 vs. 2020; *p* = 0.013]).

## 4. Discussion

Although significant progress has been made in developing diagnostics [27,28,29,30,31], vaccines [29,32,33,34], and treatment protocols [3,35,36,37,38] for canine parvovirus (CPV) since its emergence, few methodologies have been developed to attempt to limit or slow the spread of this disease within vulnerable populations. This may be partially due to a belief that reductions like this are not possible, given the extensive period during which CPV can survive in the environment [4] and difficulty in disinfecting CPV-infected areas (i.e., dog parks, homes, shelters, etc.) [4,15]. However, if population-level human behavior can be linked to infection rates, preventative methods to reduce the risk of infection in vulnerable populations beyond the typical vaccination recommendations [39,40] might be developed and/or more widely adopted. In the present study, we take advantage of the extra-ordinary circumstances provided by the COVID-19 pandemic to examine changes in CPV population in a mass-treatment intensive care unit (ICU) dedicated exclusively to the treatment of CPV. We observed a substantial drop in CPV patients immediately after the state-wide “reopening” event on 01 May 2020, with a minimum on 11 May 2020 when viewed in a two-week moving average. This finding may indicate an overall reduction in infection rates or a delay in the typical CPV season (which was still observed in these data in June, though not in the typical peak month of May [3]). In either case, its coincidence with large-scale changes in human behavior associated with the COVID-19 pandemic suggest that this reduction in CPV infections is related to human behavior and, therefore, may be employed outside of the context of a pandemic for the purposes of reducing CPV infection.

In addition to the typical on-site treatment described in this work, Austin Pets Alive! offers an at-home protocol [3] to individuals who do not wish to give up their animal but cannot afford care by a veterinarian. This option is only given to owners whose animals are not critically ill at the time of evaluation (i.e., do not present with bloody diarrhea, excessive vomiting, inappetence, and lethargy). If increases in this at-home care were seen in May of 2020, this could explain the dip in CPV-infected patients that was observed. However, no such increase was observed (see Figure 1). It is interesting to note that as the 2020 CPV season began in the month of June, a large increase in at-home treatment cases was observed, which may represent a continuation of the effects of the pandemic.

Despite evidence suggesting that reduced CPV infections were related to widespread changes in human behavior during the “stay-at-home” mandate, several potential alternative explanations warrant evaluation and discussion. Although Austin Pets Alive! changed no policies regarding the intake of animals and did not reject animals with CPV infections at any point during the periods in question, we investigated the possibility that changes at other shelters in Central Texas were related to our results. We examined potential differences in geographical animal sourcing at APA! from other shelters across Texas during 2020 which might indicate changes in policy at these shelters; however, no differences were found (see “Shelter intake policy and geographical sourcing”). Of course, this does not rule out the possibility that a failure to seek care in general drove this difference, a possibility that is extraordinarily difficult to rule out in general. However, if failure to seek care drove the differences that were observed, it might be expected that the dip would persist into peak CPV season or start in a time-locked manner to the drastic increase in unemployment (or other macroeconomic factors) seen in April in Texas and the Austin area [41]. Thus, the transient nature of the observed effect reduces the likelihood that failure to seek care is driving the effect, though it does not eliminate this possibility.

Another alternative explanation for our results could be that changes in breeding practices or animal acquisition rates across the period analyzed led to an overall decrease in the vulnerable population available for CPV infection. This is also difficult to assess, though future studies may examine if there was a decrease in the number of dogs born in the first months of 2020. However, when accounting for the typical two-month gestation period [42] and the notion that maternal antibodies may provide protection for a short period after birth [43,44,45,46,47,48], this explanation leaves little time for the onset of COVID-19 policies in the United States (mid-March 2020) to produce a maximum effect of decreased CPV-vulnerable populations in mid-May 2020, as observed. Similarly, reductions in pet acquisitions in the early portions of the pandemic could potentially explain a decrease in CPV infections, though further studies would need to examine this more holistically. During the period of 01 March 2020 to 01 June 2020, on balance, Austin Pets Alive! and Austin Animal Center adopted out more animals than the previous year, making this explanation potentially less convincing [49].

Finally, it is possible that weather patterns could have contributed to the observed decrease in CPV infections. As far as the authors are aware, no significant weather events or natural disasters (such as hurricanes or fires) occurred during the period in question in Central Texas or immediately preceding it. The distribution of intakes from various geographic regions did not change over the period in question, indicating that if weather can be used to explain the effect, the weather pattern would have to span nearly the entirety of Central, South, and East Texas (a region approximately 430,000 square kilometers in area, or somewhere between the size of Sweden and the size of Spain). Future investigations may seek to determine the degree to which temperature and precipitation may relate to infection rates. However, it is noteworthy that even if this association can be established, this could still be due to changes these weather patterns cause in human activity.

The ecological study of CPV infection has many limitations, enumerated above, and cannot conclusively show that CPV infection rates are being modified by human activity. However, the present data analysis takes advantage of a generationally-defining event in the COVID-19 pandemic, combined with seven years of high-resolution historical data on CPV intakes at the largest CPV facility in Texas, to provide a potential correlation between human activity and CPV incidence. We hope the relationship uncovered herein inspires the pursuit of additional research which attempts to directly intervene in human behavior to reduce CPV infections. Future studies could attempt targeted interventions, potentially making use of the following practices: (1) enforcing age restrictions on high-risk CPV areas such as dog parks (to ultimately reduce CPV exposure for dogs too young to be fully vaccinated), (2) performing periodic testing of public spaces such as parks and waterways in which dogs are likely to be present, and/or (3) increased public education regarding regions at a high risk for propagating CPV infections and/or regions that have recently tested positive. In addition to these interventions, information campaigns educating the public on the importance of vaccinations and protecting unvaccinated, under-vaccinated, or incompletely vaccinated dogs may help owners make more informed choices on behalf of their pets, which could enable them to help avoid their pets contracting CPV.

## Figures and Tables

**Figure 1 viruses-12-01419-f001:**
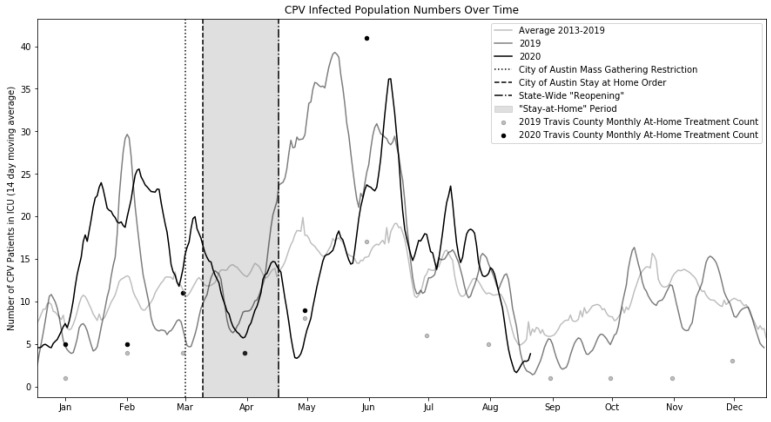
Average canine parvovirus (CPV)-infected animals over time. This figure, presented solely to illustrate the raw data under consideration in this work, shows the average population of CPV-infected animals in the intensive care unit (ICU) during 2019 and 2020 as well as the historical average ICU population from 2013–2019. The period of “stay-at-home” orders which lasted from 24 March 2020 until the Texas state-wide “reopening” on 01 May 2020 is highlighted (gray shading). Note the large difference between the 2019 and 2020 data which occurs immediately following the state-wide “reopening”. Additionally, the monthly count of at-home treatment animals (all Travis County sourced) is shown for 2019 and 2020, and, although no difference is seen during the low period in May, a large upswing in at-home care patients is seen as CPV season begins. All data are presented in two-week moving averages.

**Figure 2 viruses-12-01419-f002:**
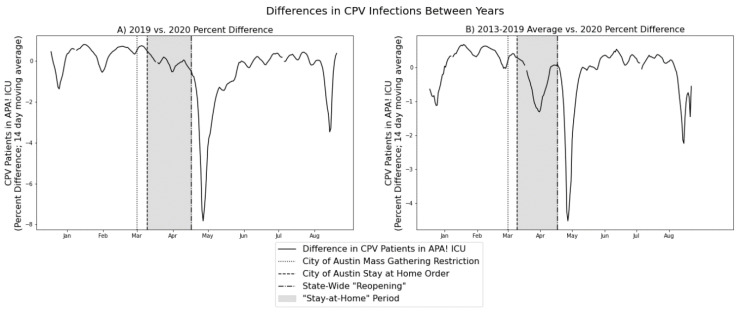
Differences in CPV infections between years. This figure contains the time series differences in CPV infections between 2019 and 2020 (**A**), as well as between the average of 2013–2019 and 2020 (**B**). Importantly, although a decrease in the CPV infection difference can be observed during the pandemic-related “stay-at-home” order (gray shading) when compared to the historical trend (**B**), this same decrease was indeed present in 2019 (**A**; also see Figure 1). Therefore, this decrease is unlikely to be related to changes in human behavior due to the “stay-at-home” order. The larger decrease, however, immediately after the “reopening” event, is not seen in any previous years. The associated date of maximum difference was 11 May 2020, 11 days after the state-wide “reopening”. All data are presented in two-week moving averages.

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
