# Peer review of "Changes in Mass Treatment of the Canine Parvovirus ICU Population in Relation to Public Policy Changes during the COVID-19 Pandemic"

_viruses, 2020, doi:10.3390/v12121419_

Round 1

Reviewer 1 Report

The manuscript explored the patterns of CPV prevalence in domestic dogs pre, during and post the ‘stay-at-home’ period in Texas due to CoVid-19. The authors examined infection data from Austin Pets Alive! and found evidence of changes in prevalence likely promoted by changes in human behaviour/activity levels during this time.  

I found the study very interesting, well written and nice to read. It was great to see how the pandemic is impacting other sectors of our society and also seeing these types of data being used.

My main concern is related to the data analysis. From the data shown I believe the impact of the stay-at-home period on CPV infection will be demonstrated and the current analysis might be correct, however the methods are not very clearly described and so its appropriateness and reasoning are not clear to me and so seem weakly supported.

How was the statistical comparison for Figure 1 done? The kernel density estimates determine whether there were differences but not the magnitude of the difference not when these might have happened.Also, how are the analysis taking into account temporal variability/seasonality?

I would think that something like detrending the time-series and then a glm type analysis or using an ARIMA to investigate whether there were statistical differences would be more informative. Alternatively, there are also tests such as granger tests that can be used to show predictability. Could show that historically there was predictability of CPV but was lost in 2020. To strengthen the analysis, it would be good to show consistency in historical years before comparing to 2020. Averages across years hide small patterns that may not be so different to current small changes, especially since from observation of fig 1 it seems that 2020 was already slightly different from 2019 before restrictions.

Fig. 2: The main difference in CPV cases seems to happen just before (e.g. in A) or simultaneous (e.g. in C) the peak of Covid-19 cases. How was this formally tested and if they happen prior to Covid, can it be claimed to be Covid related?

Fig. 3: Were the data included here restricted to the months when 2020 data is available? If not, could this bias the estimates?  

Finally, for the geographical distribution analysis, what the variables going into these? assuming it is proportion of CPV cases per year/county, can these really be considered dichotomous instead of continuous? This is a key assumption for the McNemmar test used.

In addition to these clarification to the analysis, the introduction could clarify better when and why CPV infection occurs. For example, what is the vaccination coverage, are there seasonality, the authors mention CPV cycle in wildlife - what type of wildlife and does this contribute to domestic dog CPV and what human behaviours typically contribute to CPV in dogs.

In the discussion, the authors mentioned a decrease in animal acquisition during the pandemic. Are there data for this? I keep hearing that dog adoptions and consequently breeding increased during this time so seeing data to support either side would

Author Response

(See attached pdf for these same comments in case formatting doesn't come through)

Thank you so much for your thoughtful review. We appreciate your suggestions to improve our manuscript and have implemented a variety of changes in order to address your concerns. Based on your feedback, we improved the presentation and analysis of what you noted to be our key finding, the difference in CPV population related to the pandemic-related public policy changes. We appreciate your guidance and want to ensure our results are clearly examined and presented, so if we have failed to address any of the methodological concerns you have raised, please let us know if there are any particular statistical approaches you think would be more appropriate.

The manuscript explored the patterns of CPV prevalence in domestic dogs pre, during and post the ‘stay-at-home’ period in Texas due to CoVid-19. The authors examined infection data from Austin Pets Alive! and found evidence of changes in prevalence likely promoted by changes in human behaviour/activity levels during this time.  

I found the study very interesting, well written and nice to read. It was great to see how the pandemic is impacting other sectors of our society and also seeing these types of data being used.

My main concern is related to the data analysis. From the data shown I believe the impact of the stay-at-home period on CPV infection will be demonstrated and the current analysis might be correct, however the methods are not very clearly described and so its appropriateness and reasoning are not clear to me and so seem weakly supported.

We significantly clarified the methods through the paper, simplifying and removing unneeded components, and modifying some statistical tests to try to address these concerns. Please see the below responses to the specific feedback for details. We hope these modifications make the results clear.

How was the statistical comparison for Figure 1 done? The kernel density estimates determine whether there were differences but not the magnitude of the difference not when these might have happened.

Figure 1 does not contain any statistical analyses and is presented purely for illustration of the data being analyzed so that interested readers can visually inspect the patterns under discussion. To clarify that the first figure is intended to allow for visual examination of the data, not for statistical analysis, we modified the first sentence of the figure caption to read as follows (bold denotes new text):
“This figure, presented to illustrate the raw data under consideration in this work, contains the 14-day moving average population of CPV infected animals in the ICU during 2019 and 2020 as well as the historical average population from 2013-2019.”

Also, how are the analysis taking into account temporal variability/seasonality?

Previous work (https://pubmed.ncbi.nlm.nih.gov/32485882/) found seasonal trends in these data, and, as noted in the abstract and in various portions of the manuscript, the typical seasonal peak would occur in precisely the month we observe the drastic reduction in cases. These temporal patterns were not directly addressed in the paper, though they are implicitly considered by differencing identical dates across years. 

I would think that something like detrending the time-series and then a glm type analysis or using an ARIMA to investigate whether there were statistical differences would be more informative. Alternatively, there are also tests such as granger tests that can be used to show predictability. Could show that historically there was predictability of CPV but was lost in 2020. To strengthen the analysis, it would be good to show consistency in historical years before comparing to 2020. Averages across years hide small patterns that may not be so different to current small changes, especially since from observation of fig 1 it seems that 2020 was already slightly different from 2019 before restrictions.

This is a very interesting suggestion and an approach we considered. Previous work by us using these data (excluding the currently considered period in 2020) did perform a seasonal decomposition analysis via a bayesian structural time series approach in the Prophet software package to attempt to address some season components. The MAPE values for these models were fairly poor (~25% with a 7 day forecast horizon backtest) due to substantial unexplained variability in the signal. This may, in part, be driven by the fact that animals often come into the ICU in litters, resulting in large spikes which were, in fact, associated with a single infection event; though we do not have the data to address this hypothesis at this time. 

The analyses presented in this work were designed to be as approachable as possible by an audience with limited statistical comfort to illustrate what we believe is a fairly straightforward reduction in cases, time locked to significant social changes. Though we hope future work can attempt to perform more complex multivariate analyses of these time series data.

Fig. 2: The main difference in CPV cases seems to happen just before (e.g. in A) or simultaneous (e.g. in C) the peak of Covid-19 cases. How was this formally tested and if they happen prior to Covid, can it be claimed to be Covid related?

Thank you for bringing our attention to this opportunity to make our data presentation more clear. We did not directly test the relationship between Covid-19 cases, and are therefore not claiming the decrease in CPV cases are directly related to Covd-19 cases. Instead, we provide support for the hypothesis that changes in human behavior based on mandated lockdown due to the Covid-19 pandemic alters CPV cases. In order to help avoid this confusion with readers, we removed the visualization of COVID cases from Figure 2.  They were originally provided as a point of interest, but they are not directly related to the findings. In addition, we changed the title to Changes in Mass-Treatment Canine Parvovirus ICU Population Related to Public Policy Changes During the COVID-19 Pandemic (bolding indicates new text).

To further comment on the above question, we believe the relationship between the human covid cases and the CPV infections is indirect. There are two reasons for this, as outlined in the first paragraph of the discussion and the caption of figure 2. First, the relationship to covid cases is likely coincidental in part because previous work shows parvo season occurs in May and June (our finding was that May did not show a population rise associated with parvo season as expected). It just so happens this is also when COVID cases began to rise in the region, but an increase in cases of both are being driven by different factors. However, the reduction in CPV cases from what is typically seen was temporally related to the major social changes preceding the typical parvo season. So the timeline, as we would describe it, is: 1) covid stay-at-home orders drastically alter the behavior of humans in the months before parvo season would typically begin, 2) changes in human behavior result in a change in CPV infection, coincidentally right as parvo season should begin (and, potentially, a delay in the onset of major covid case outbreaks; though this is outside the scope of this work), 3) covid cases spike after stay-at-home measures are removed. Critically, it is not the increase in number of individuals infected with covid which lead to a decrease in parvo, but the changes in human behavior driven by city and state-wide orders which did, in fact, precede the effects being discussed in this work. 

Fig. 3: Were the data included here restricted to the months when 2020 data is available? If not, could this bias the estimates?  

Other reviewers expressed concern over the length of the paper, so we removed this figure. However, because the nature of the analysis is that it was a plot of the distribution of the differences between 2020 and prior years (locked to each day), months where data was not yet available in 2020 were implicitly removed as the difference is undefined. Including months for which we do not have 2020 data would require changing the nature of the analysis from a distribution of differences to a difference of distributions which, per previous discussions, would require a much more significant examination of the temporally-dependent properties of the signal. 

Finally, for the geographical distribution analysis, what the variables going into these? assuming it is proportion of CPV cases per year/county, can these really be considered dichotomous instead of continuous? This is a key assumption for the McNemmar test used.

Based on your feedback, we reconsidered the strategy of analysis for these data. The variables in question are as follows: The dichotomous variable in question are the years (2020 and 2017-2019), not the number of CPV intakes per county per year. The number of CPV intakes per county per year is the dependent variable, which are indeed continuous, but not required to be dichotomous per the McNemar test assumptions.  

After careful reconsideration of these data, we ran our analysis with the McNemar-Bowker test which more accurately computes the degrees of freedom given the larger contingency (i.e. non-2x2) and handles the proportional nature of the independent variable more effectively. If we only had 2 counties in question, we believe the original McNemar test would have been appropriate, but given the number of counties is large, we believe the Bowker symmetry extension of this test more appropriately addresses the situation. No difference in the result was seen from this change (i.e. there is no significant difference in geographic source distribution of CPV animals between 2020 and 2017-2019). If the reviewer has a particular test in mind they would like to see to address the question of whether there were differences in geographic sourcing between 2020 and 2017-2019, we would be happy to run it. We certainly want to ensure we are addressing this potential explanation of changes in CPV intakes effectively.

In addition, based on another reviewer’s feedback, we have chosen to de-emphasize this result (as it was a null finding) by removing the figure associated with it and moving the finding to the methods section to indicate that this was checked, but it was not considered part of the key findings (see section 2.2 which reads: 

“Austin Pets Alive!’s Parvo ICU intake practices did not change during any of the periods in question (i.e. no artificial reduction in population due to policy was present). In addition, no changes in geographic distribution of animal intake sources at APA! were observed, as measured by a McNemar-Bowker test for Multiple Correlation Proportions (2017-2019 vs. 2020; χ2=9.48, ν=528, p~1).”).

In addition to these clarifications to the analysis, the introduction could clarify better when and why CPV infection occurs. For example, what is the vaccination coverage, are there seasonality, the authors mention CPV cycle in wildlife - what type of wildlife and does this contribute to domestic dog CPV and what human behaviours typically contribute to CPV in dogs.

Although we do understand that CPV spreads via the fecal oral route, we do not fully understand the distribution of possible environmental factors which contribute to CPV infection. We have elaborated on this point in the introduction by highlighting 2 key factors which are fairly well understood:

  1. CPV can survive for significant periods in the environment (i.e. on surfaces such as grass and pavement)
  2. Wildlife can be infected with CPV, perpetuating the disease in the environment

Unfortunately, we do not have data on the vaccination coverage for CPV. Please let us know if you have any further clarifications or citations we can provide on this point. The section in the introduction in question now reads:

“Seasonal effects in CPV have been observed worldwide [6–10]. In Central Texas, seasonal trends in incidence have been observed during the past decade, specifically peaking in the late spring / early summer months of May and June. Although much is known about the mode of transmission (i.e. viral spread via the fecal-oral route; [11]), and it is standard practice in Central Texas animal shelters to inoculate against for CPV using the highly effective and widely available CPV vaccine, significant natural disease reservoirs must persist which perpetuate the disease among domesticated dogs. These reservoirs may exist in the form of wildlife (large cats, racoons, racoon dogs, arctic foxes, and mink; though evidence of transmission via this route is unclear; [12–14]) and/or unvaccinated feral animal populations. Additionally, surfaces in the environment such as grass or pavement may house infections, given CPV’s ability to survive on surfaces for extended periods of time, unless significant disinfection procedures are implemented due to the hardy nature of this nonenveloped virus [4,15]. ”

In the discussion, the authors mentioned a decrease in animal acquisition during the pandemic. Are there data for this? I keep hearing that dog adoptions and consequently breeding increased during this time so seeing data to support either side would

Within the scope of the largest geographic region in the study (i.e. Travis county, in which the Parvo ICU resides), this is called out in the third paragraph of the discussion: “During the period of 01 March to 01 June , on balance, APA! and AAC adopted out more animals than last year, making this explanation potentially less convincing [24].”. Unfortunately, more extensive evaluation of acquisitions are beyond the scope of this work (and data is not available from all of the sourcing organizations in this study), but we agree that this is a fascinating topic and look forward to future work that addresses changes in acquisition and breeding, directly. If the reviewer has any citations they would like to suggest which might address these points, we would love to include them.

Reviewer 2 Report

This is a well-written paper with one convincing observation namely that the ongoing pandemic coincides with changes the incidence of canaine parvovirus infections in Texas. Year 2020 incindences were compared with those from 2019 and with an average from the period between 2013-2019.The differences is convincingly demonstrated. The second theme of the paper is that this knowledge may benefit future This attempts to fight the disease. This is not obvious from the data provided. Numerous changes in human behavior, which have taken place during the lock down, are conceivable explanations as the authors point out in the discussion.

My conclusion is that the scientific value of the paper is very limited and does not warrant a full-length publication with many figures. Scientists in the field are likely to be interested in the reported observation. Therefore, a note describing the observation including one figure could make sense.

Author Response

This is a well-written paper with one convincing observation namely that the ongoing pandemic coincides with changes the incidence of canaine parvovirus infections in Texas. Year 2020 incindences were compared with those from 2019 and with an average from the period between 2013-2019.The differences is convincingly demonstrated. The second theme of the paper is that this knowledge may benefit future This attempts to fight the disease. This is not obvious from the data provided. Numerous changes in human behavior, which have taken place during the lock down, are conceivable explanations as the authors point out in the discussion.

My conclusion is that the scientific value of the paper is very limited and does not warrant a full-length publication with many figures. Scientists in the field are likely to be interested in the reported observation. Therefore, a note describing the observation including one figure could make sense.

Thank you so much for your helpful feedback on this work; it is much appreciated. We agree that the length of the originally submitted work was not proportional to the scientific value it contributes. In order to try to bring these qualities into better alignment, we have removed the Appendix, Figure 3, and Figure 4. Additionally, we simplified Figure 2 and its associated explanations to highlight our key data. In response to other reviewer suggestions, we modified some statistical methods slightly to more effectively address the phenomena in question. This has resulted in a 40% reduction in the paper size. In addition, we have asked the editor if it should be considered for a different publication type which may better reflect the findings.

With regard to fighting CPV: We agree that these data do not directly provide tools which might be used to combat CPV infections and outbreaks; however, we hope the relationship between this societal-scale, human-behavior changing event (covid 19) and changes in CPV infections lead to additional research which attempts to directly intervene in human behavior to reduce CPV infections. Whether the proposed interventions in the final paragraph of the paper will be successful is unknown, but we hope this small piece of evidence suggesting a link between human behavior and CPV infection can be used to motivate future work on these, and other, interventions.

Round 2

Reviewer 1 Report

The authors addressed most of my comments satisfactorily. The current manuscript is almost half the size of the original version and does not warrant publication as full article - however, I believe the plan now is to publish as a note, for which I agree the content is appropriate and support publication in present form.

I would have still liked to see a comparison of CPV among previous years to ensure that the difference seen in 2020 is indeed real and not present among other years, but for a short note I feel this is not strictly necessary. 

Reviewer 2 Report

The authors have changed the manuscript in line with my recommendations. Thus I am willing to recommend publication.